# LBRT: Local-Information-Refined Transformer for Image Copy–Move Forgery Detection

**DOI:** 10.3390/s24134143

**Published:** 2024-06-26

**Authors:** Peng Liang, Ziyuan Li, Hang Tu, Huimin Zhao

**Affiliations:** School of Computer Science, Guangdong Polytechnic Normal University, Guangzhou 510630, China; lzy4088@126.com (Z.L.); thlovepc@126.com (H.T.);

**Keywords:** image copy–move forgery detection, self-attention, transformer, local information refinement

## Abstract

The current deep learning methods for copy–move forgery detection (CMFD) are mostly based on deep convolutional neural networks, which frequently discard a large amount of detail information throughout convolutional feature extraction and have poor long-range information extraction capabilities. The Transformer structure is adept at modeling global context information, but the patch-wise self-attention calculation still neglects the extraction of details in local regions that have been tampered with. A local-information-refined dual-branch network, LBRT (Local Branch Refinement Transformer), is designed in this study. It performs Transformer encoding on the global patches segmented from the image and local patches re-segmented from the global patches using a global modeling branch and a local refinement branch, respectively. The self-attention features from both branches are precisely fused, and the fused feature map is then up-sampled and decoded. Therefore, LBRT considers both global semantic information modeling and local detail information refinement. The experimental results show that LBRT outperforms several state-of-the-art CMFD methods on the USCISI dataset, CASIA CMFD dataset, and DEFACTO CMFD dataset.

## 1. Introduction

The increasing popularity of image editing software has enabled people to create a large number of realistic forged images at a low cost. Maliciously manipulated images can have many negative consequences: they can be used for fraud, to spread fake news, to fabricate false proof, and to manipulate public opinion. Therefore, it is necessary to research effective methods for the detection of image manipulation.

In the context of image content manipulation, copy–move forgery [1] refers to the copying of a region from an image and its pasting onto another location within the same image. Figure 1 shows two examples of content manipulation through the copy–move operation. The purpose of copy–move forgery detection (CMFD) is to detect copy–move forgery in an image and to locate the regions that have been tampered with at the pixel level. Because copy–move forgery takes place within the same image, the copied and pasted regions only exhibit minor variance in details such as edge artifacts, and there are few useful clues in terms of the frequency domain or depth features, which makes CMFD a difficult task.

CMFD is closely tied to the image’s global contextual and semantic information as well as attention to the details of the fabricated edges. As a result, both global semantic features and local edge features have to be considered. Deep learning methods based on deep convolutional neural networks (DCNNs) have recently emerged as a research hotspot due to benefits such as fewer hyperparameters, high generality, and a strong capacity to learn semantic features through convolution [2,3,4,5,6]. A DCNN employs convolutional modules to extract low-level features such as contours, edges, and textures. With the expansion of its receptive field, it progressively captures semantic-aware high-level features. However, as the image size is continuously reduced via convolution, many features, particularly those of small objects, will unavoidably be lost. The network’s receptive field is limited by the size of the convolution kernel, making it difficult to collect adequate long-range dependency information.

The current advancements in deep learning approaches for copy–move forgery localization tend to use a self-correlation computation module after feature extraction [2,4,5,7]. The self-correlation computation module calculates the feature similarity between the pixels representing a region in the feature map, and it uses the feature map of the similarity scores for decoding. It aims to enhance the representation of long-range dependency information. However, self-correlation computation can only partially resolve the deficiencies of DCNNs in extracting global similarity information. It still overlooks the issue of the loss of detailed information about local forgery artifacts, which can lead to the mislabeling of non-manipulated objects that are semantically or structurally similar to manipulated areas, thus limiting the model’s localization performance. The current approaches also incorporate designs such as Atrous Spatial Pyramid Pooling (ASPP) [4,5,7], dense extraction networks [3], and U-Net [7] to attempt to preserve more semantic information at various scales in the feature map. However, these designs still fall short in adequately retaining the details of local manipulation traces, resulting in incomplete contours in the forgery localization mask and the inability to identify manipulation of non-foreground objects.

Transformer’s encoding structure [8] provides an alternative encoding method to overcome the limitations of convolution, allowing features to be encoded without decreasing the image resolution. Instead, it performs direct sequence-to-sequence self-attention calculation on the patches, searching for long-range contextual information from the beginning of the feature extraction process, which is conducive for the extraction of information about the similarity between copied regions and pasted regions in CMFD tasks. However, the use of Transformer for CMFD tasks that need fine-grained segmentation still faces two important obstacles. Firstly, due to computational restrictions, the images are commonly divided into large-sized patches in the basic Transformer encoder. Self-attention is computed between large coarse-grained patches, whereas the processing and encoding of particular pixels inside the image blocks are ignored. This is incompatible with the learning of delicate edges within patches, which is required for CMFD tasks in order to collect the local information in the regions that have been tampered with. Secondly, Transformer’s performance is strongly dependent on large-scale pre-training, and the network necessitates a vast number of parameters. We intend to improve the Transformer structure with only a minor parameter increase while eliminating the requirement for large-scale pre-training. CNN-T GAN [9] is a pioneering model that utilizes Transformer for feature extraction in the CMFD task. However, CNN-T GAN does not apply pre-training weights in the Transformer branch due to its network structure’s design, leading to poor ability to extract global information. In reality, the novel loss functions devised in CNN-T GAN contribute significantly to enhancing the localization performance.

We propose a method for the detection and localization of copy–move forgery based on a dual-branch Transformer structure to address the limitations of the standard Transformer structure in extracting local information within blocks. Our method is called the Local Branch Refinement Transformer (LBRT). LBRT consists of a global context modeling branch and a local refinement branch. The global context modeling branch makes use of the Transformer encoder’s outstanding modeling capabilities to capture long-range contextual information via self-attention calculations between image patches. It focuses on identifying the copy–move regions that have been tampered with by capturing the dependencies and correlations between distinct spatial regions in images via a global receptive field. The local refinement branch, on the other hand, improves the Transformer encoder by conducting further fine-grained partitioning within the image patches generated from the preceding branch, namely via an Intra-Patch Re-Dividing Layer (IPRL). This allows the network to pay attention to the correlations between different local spatial regions within the global patches, optimizing the detection of forged region edges. Following the separate extraction of global and local information, the features from both branches are fed into a fusion module, which effectively combines the two types of features, optimizing the details of the global features with the local features. Finally, the fused feature map is fed into a designed decoder to predict a mask that indicates the locations where copy–move forgery has occurred.

In summary, the main contributions of this work are as follows.

A novel CMFD network based on Transformer is proposed, which utilizes a dual-branch structure to extract both long-range global information and fine-grained local information. LBRT addresses the limitations of previous DCNN-based CMFD networks in extracting global information and enhances the ability of the Transformer encoding structure to extract local information. It leverages local edge information to refine the global features, thereby improving the accuracy of locating copy–move forgery regions.Transformer encoding is refined in the local branch in accordance with the characteristic of local information extraction. The Intra-Patch Re-Dividing Layer (IPRL) in the local branch not only ensures the proper extraction of local features for each image patch obtained from the global branch but also realizes the addition of local information with only a minor increase in the number of model parameters. It is not necessary to repeat large-scale pre-training.Extensive experiments on the USCISI, CASIA CMFD, and DEFACTO CMFD datasets demonstrate that the proposed network outperforms advanced techniques in the CMFD field, including both traditional DCNN models and DCNN models with additional attention mechanisms.

## 2. Related Work

### 2.1. Traditional CMFD Methods

Currently, CMFD methods generally follow three basic steps [10]: feature extraction, feature similarity matching, and post-processing. With the advancement of research, the currently dominant CMFD methods can be divided into two categories: traditional methods and deep learning methods.

Traditional methods include block-based and keypoint-based methods. Block-based methods, such as Zernike moments [11], the Discrete Cosine Transform (DCT) [12], Principal Component Analysis (PCA) features [13], and the Discrete Wavelet Transform (DWT) [14], divide an image into overlapping or non-overlapping blocks and compare the feature similarity between them. Keypoint-based methods, including the Scale-Invariant Feature Transform (SIFT) [15], Speed Up Robust Features (SURF) [16], Oriented FAST and Rotated BRIEF (ORB) [17], and Fast Retina Keypoint (FREAK) [18], extract keypoints from the entire image and use them to perform feature description and matching. Block-based methods are simple to implement but have poor robustness to wide-ranging rotation and scaling. Keypoint-based methods are more robust to geometric transformations such as rotation and scaling, but they struggle to extract sufficient keypoints for smooth, highly deformed, or small regions. Both types of traditional methods achieve the accurate detection of manipulated regions, but they require manual adjustment of numerous parameters, and each module needs to be optimized separately, leading to poor generalization capabilities.

### 2.2. DCNN-Based CMFD Methods

Deep learning CMFD methods mostly rely on DCNNs, which can accomplish the joint optimization of modules through an end-to-end design and do not require the manual adjustment of many parameters. These methods typically follow an encoder–decoder framework consisting of three main modules: a CNN feature extractor, a feature matching module, and a decoder. These modules are responsible for feature extraction by calculating the feature similarity and obtaining the tampering mask, respectively. BusterNet [2] adopts a dual-branch design, utilizing a simple CNN feature extractor to extract copy–move manipulation features in the manipulation branch while adding a self-correlation calculation module in the similarity branch to determine the similarity of regions that have been tampered with, which thereby enables differentiation between the source and target regions. Dense-InceptionNet [3] integrates the architectures of DenseNet [19] and InceptionNet [20] to extract multi-dimensional and multi-scale dense features and computes the correlations of the dense features using an enhanced feature correlation matching module. Some DCNN methods also incorporate other deep learning methods, such as Generative Adversarial Networks (GANs) [21] and attention mechanisms [22], to further improve their CMFD models. CMSDNet and STRDNet [4] enhance BusterNet with a single-branch dual-network design for similarity detection and source/target discrimination, and they use channel attention, spatial attention, and spatial pyramid pooling to enhance the similarity detection network. DOA-GAN [5] introduces dual-order attention to better capture the position information of copy–move objects and discriminative feature information, and it utilizes a GAN to further obtain more accurate localization results. UCM-Net [7] considers the CMFD task as a particular type of semantic segmentation task, introduces a U-Net-like CMFD framework that processes large and small regions that have been tampered with differently, and exploits multiple U-shaped residual U-block modules to capture both global and local information.

In addition to dedicated CMFD solutions, there have been remarkable DCNN-based image forgery detection schemes in recent years. These networks can detect and localize various types of image content forgery, including copy–move forgery. DenseFCN [6] adopts the Full Convolutional Network (FCN) [23] framework and DenseNet [19] structure for feature extraction and decoding. MVSS-Net [24] addresses the sensitivity of manipulation in novel data and the specificity of manipulation via multi-view feature learning and multi-scale supervision. Meanwhile, PSCC-Net [25] adopts a more complex design, as it proposes a progressive spatio–channel correlation network to detect and localize the manipulation of image content. MVSS-Net and PSCC-Net both incorporate spatial attention and channel attention. Although none of these methods have self-correlation calculation modules, MVSS-Net and PSCC-Net generally outperform DenseFCN in image content forgery detection tasks, especially regarding copy–move forgery.

The DCNN network performs well on smaller datasets due to its inductive biases, such as translational equivariance and locality. The convolutional down-sampling operation also reduces the computational cost of feature correlation. However, compared to traditional methods, the DCNN network has lower accuracy at identifying regions that have been tampered with and especially at detecting small objects. There is also significant room for improvement in its robustness to various attacks and its ability to differentiate between source and target regions.

### 2.3. Transformer Methods

Transformer is a network structure that is widely used in natural language processing (NLP) due to its unparalleled computational efficiency and scalability. ViT [26] was the first to apply the standard Transformer structure to computer vision (CV). It demonstrated that although the self-attention mechanism loses the inductive bias specific to the DCNN, the performance of the basic Transformer for image classification tasks can still match or even surpass that of DCNNs, especially with the support of large-range pre-training. Subsequent research has also shown that self-attention and Transformer can achieve outstanding performance in other CV tasks, such as image classification (DeiT [27], Swin Transformer [28], PVT [29], TNT [30]), object detection (DETR [31], and Deformable DETR [32]) and image segmentation (SETR [33] and TransUNet [34]), effectively helping networks to focus on category or instance features with a global perspective. In image forgery detection, ObjectFormer [35] comprises a CNN–Transformer hybrid structure network; it extracts RGB and frequency-domain features via a DCNN backbone and detects various types of image content forgery via the Transformer structure based on object queries and cross-attention. CNN-T GAN [9] comprises a CNN–Transformer-based network, too, and it utilizes a GAN to improve the network’s ability to discriminate between source/target/untampered regions.

Among the numerous improvements to Transformer, TNT [30] incorporates the visual word embeddings calculated by the inner blocks into the visual sentence embeddings calculated by the outer blocks, providing local information enhancement for Transformer that allows it to excel at encoding global information. Although this improvement does not significantly increase the number of parameters, it requires large-scale pre-training as it involves modifying the backbone network. EDTER [36] proposes a two-stage structure based on Transformer to enable edge detection. Its dual-branch design and the Non-Overlapping Sliding Window (NOSW) avoid modification to the backbone and enable the network to capture more local edge features, but the method of partitioning local windows is equivalent to evenly distributing the image to multiple Transformer encoders for fine-grained self-attention calculations, which still fails to fully capture the similarity between different spatial regions within each block. The local branches based on NOSW also result in a large increase in the number of parameters, and the phased training method almost doubles the training time.

## 3. Methodology

Inspired by the visual word embeddings designed in TNT [30] and the dual-branch structure in EDTER [36], we propose a local information-enhanced dual-branch Transformer network for CMFD tasks named the Local Branch Refinement Transformer (LBRT), which ensures that the local branch’s patch segmentation is based on the segmentation of small patches from large patches in the global branch, making the detail features within each patch able to be extracted. Meanwhile, LBRT achieves the extraction of local information and the refinement of local details in the global information, while adding very few parameters. Moreover, pre-trained weights are not needed for the local branch’s initialization, and joint training with the global branch shortens the overall training time.

As shown in Figure 2, our network follows an overall encoder–decoder structure. Given an input image X, we first apply a ViT-base [26] encoder with a dual-branch design to extract global similarity features $F_{g}$ and local detail features $F_{l}$ for copy–move forgery. The feature extraction process consists of the global context modeling branch and the local refinement branch. The global branch aims to extract features with a global perspective, enabling the model to focus on the source/target regions where copy–move manipulations occur in the image. The local branch aims to extract features within smaller image patch dimensions, allowing the model to focus on the edge details of local regions. Following the feature extraction stage, a fusion module combines the features $F_{g}$ derived by the global branch and $F_{l}$ derived by the local branch, producing a final feature representation $F_{\mathrm{gl}}$ that incorporates both the similarity of long-range regions that have been tampered with and detailed information from local regions. After feature fusion is finished, the 2D feature map $F_{\mathrm{gl}}$ is sent to the decoder to ultimately obtain a copy–move tampering mask Y.

### 3.1. Global Context Modeling Branch

Figure 3 depicts the specific structure of the global context modeling branch. Using an encoder with ViT-base standards [26] for feature extraction, the model is able to extract two correlated regions from the original image, namely the source and target regions of the copy–move forgery. The ViT-base encoder has the advantage of being more in line with the characteristics of the CMFD task. The core mechanism in the ViT-base encoder, called global self-attention calculation, is able to focus on the region similarity in the global view based on the global patches divided from the image. The process is analogous to the initial observation of the entire image by the human eye when detecting copy–move forgery, followed by a search for conspicuously similar regions that are subsequently included in the areas that are suspected of being tampered with. Moreover, the global self-attention calculation also replaces the self-correlation calculation module that is widely used in DCNN-based models to some extent because it has been shown to be capable of extracting the correlation between different regions of the image itself and updating the feature representation accordingly.

We denote an input image as $X\in R^{h\times w\times 3}$, where *h* and *w* are the height and width of the image, respectively. Firstly, to convert X into a 1D sequence that can be used as input for the Transformer encoder, the image has to be processed through the image pre-processing block. The image is filtered by a convolution layer with a kernel size of $p_{1}\times p_{1}$ and stride of $p_{1}$, which is equivalent to the process of flattening the image and linearly projecting it as a patch embedding $X_{p}\in R^{N_{1}\times d_{1}}$, where $X_{p}=\left[X_{p}^{1},X_{p}^{2},\ldots ,X_{p}^{N_{1}}\right]$, $X_{p}^{N_{1}}$ represents the $N_{1}$-th global patch, $N_{1}=\frac{hw}{{p_{1}}^{2}}$, and $d_{1}$ represents the channel number of $X_{p}$. Position embeddings Pos are then added to $X_{p}$ as element-wise additions, generating the global patch embedding $Z^{l}$ as the input of the global Transformer encoder, which retains the same size as $X_{p}$. The image pre-processing step can be formulated as
(1)$$Z^{l}=X_{p}+\mathrm{Pos},$$
where *l* represents the *l*-th layer of the encoder, and $Z^{l}\in R^{N_{1}\times d_{1}}$. Following the standards of the ViT-base backbone, *l*, $p_{1}$, and $d_{1}$ are set to 12, 16 and 768, respectively.

Subsequently, $Z^{l}$ is fed to the global Transformer encoder, which is composed of multiple layers of multi-head self-attention (MSA) and multi-layer perceptron (MLP) modules. The global encoding procedure enables a global perspective in order to identify regions that exhibit copy–move tampering traits throughout the image at the patch level. The MSA block and MLP layers are added before a LayerNorm layer and residual connection. Figure 4 shows the specific steps of the global Transformer encoder.

In the MSA module, $Z^{l}$ is first multiplied by three learnable matrices, $W_{q}$, $W_{k}$, and $W_{v}$, to obtain q, k, and v, respectively. Next, there are $n_{1}$ attention layers running in parallel, namely heads, within each block, which can divide the channels of q, k, and v into $n_{1}$ groups. Each group of q, k, and v is used for parallel multi-head self-attention (MSA) calculations to obtain self-attention matrix A. Then, A is multiplied by v to update the values of v based on the attention weights between different patches. The results of each head are then concatenated to obtain the final output of this module, $GMSA(Z^{l})$. The mapping of q, k, and v to the output $GMSA(Z^{l})$ is calculated as
(2)$$A=Softmax(\frac{qk^{T}}{\sqrt{D_{h}}}),$$
(3)$$GSA(Z^{l})=Av,$$
(4)$$GMSA\left(Z^{l}\right)=Concat\left(GSA_{1}\left(Z^{l}\right),GSA_{2}\left(Z^{l}\right),\ldots ,GSA_{n_{1}}\left(Z^{l}\right)\right),$$
where $q,k,v\in R^{N_{1}\times \frac{d_{1}}{n_{1}}}$. $D_{h}$ is set to 256, which is equal to the length of q, k, and v; $n_{1}$ is set to 12; GSA denotes a global self-attention calculation with the same size as q, k, and v; GMSA denotes a global multi-head self-attention calculation with the same size as the input of module $Z^{l}$; Softmax denotes an activation function implemented by softmax; Concat denotes channel concatenation.

The process of GMSA calculation involves computing the feature similarity between each global patch and the other patches using a dot product. The resulting self-attention matrix A, which represents the correlation between elements, is then used as weights for weighted summation to update the feature representation applied to each global patch. During MSA calculation, elements with a low correlation have a diminished impact, allowing the network to gradually focus on regions with high similarity, which are suspected to be regions that have been tampered with.

After performing the multi-head self-attention calculations, $GMSA(Z^{l})$ is input into the MLP module to enrich the feature representation, where the number of channels is quadrupled and projected back. The output of this module is the final output of one layer of the encoder, denoted as $Z^{l+1}$, which is also the input for the next layer of the encoder.

The final output of the global Transformer encoder, denoted as $Z^{12}$, is obtained after completing the encoding procedure for all 12 stacked layers.

### 3.2. Local Refinement Branch

It is not enough to capture the entirety of the forgery feature information by modeling the global context alone. The image is roughly divided into $p_{1}\times p_{1}$ patches in the global branch. The self-attention between the query and key is calculated based on the patches in the ensuing MSA calculations to discover the correlations between the patches, which are essential for feature extraction. Nevertheless, modeling of the local information within each patch is lacking, which may have an impact on the localization performance regarding regions that have been tampered with. Consequently, we design a local refinement branch that is implemented inside the global patches defined by the global branch to carry out further local Transformer encoding. The feature extraction procedures of two branches are run concurrently. The architecture of the local refinement branch is illustrated in Figure 5.

In the local branch, we still need to divide X into several patches, called global patches, and then flatten these patches to a 1D sequence $X_{\mathrm{sp}}\in R^{N_{1}\times p_{1}\times p_{1}\times 3}$. Note that in this branch, every global patch will be treated as an input image, i.e., during image pre-processing, every global patch will be re-divided, and each global patch $X_{\mathrm{sp},m}$ will be divided into several sub-patches with a resolution of $p_{2}\times p_{2}$, called local patches. Afterwards, each batch of local patches is sent to the linear projection layer to be processed as a local patch embedding and then added to the position embedding as the input of the local Transformer encoder $\left[z_{1}^{l},z_{2}^{l},\ldots ,z_{N_{1}}^{l}\right]$. In this case, $X_{\mathrm{sp},m}=\left[X_{\mathrm{sp},m}^{1},X_{\mathrm{sp},m}^{2},\ldots ,X_{\mathrm{sp},m}^{N_{2}}\right]$; *m* represents the *m*-th global patch; $m=1,2,\ldots ,N_{1}$; $X_{\mathrm{sp},m}^{N_{2}}$ represents the $N_{2}$-th local patch that belongs to the *m*-th global patch. Thus, pre-processing by the local branch is finished. The following procedure for obtaining the local patches is performed in the Intra-Patch Re-Dividing Layer (IPRL), and it can be formulated as
(5)$$z_{m}^{l}=X_{\mathrm{sp},m}E+\mathrm{Pos},$$
where $z_{m}^{l}\in R^{N_{2}\times d_{2}}$, which represents the local patch embedding obtained by the *m*-th global patch during image pre-processing, *l* represents the *l*-th layer of the encoder, $E\in R^{\left(3p_{2}^{2}\right)\times d_{2}}$ is the linear projection layer, $N_{2}=\frac{p_{1}^{2}}{p_{2}^{2}}$, and $d_{2}$ is the channel number of local patch embedding, which is set to a lower value of 48 in order to account for the computational cost of re-concatenating the local patch embedding after completing the following encoding process.

The IPRL design ensures that local patches are divided based on a global patch, and each batch of local patches belonging to different global patches is fed into the encoder for parallel local self-attention calculation, thereby facilitating the extraction of detailed features from each local area in the image. Thanks to the incorporation of position embedding in every local patch, the locations of all local patches in the image are recorded, which facilitates the subsequent fusion between the local and global features.

Afterwards, all local patch embeddings $\left[z_{1}^{l},z_{2}^{l},\ldots ,z_{N_{1}}^{l}\right]$ obtained from a single image are fed into the local Transformer encoder in batches. This has the same structure as the global Transformer. The local Transformer encoder calculates the self-attention within each global patch based on the local patch to find the local details within the global patches. The MSA calculation of the local branch can be formulated as
(6)$$LSA\left(z_{m}^{l}\right)=Softmax\left(\frac{qk^{T}}{\sqrt{D_{h}}}\right)v,$$
(7)$$LMSA\left(z_{m}^{l}\right)=Concat\left(LSA_{1}\left(z_{m}^{l}\right),LSA_{2}\left(z_{m}^{l}\right),\ldots ,LSA_{n_{2}}\left(z_{m}^{l}\right)\right),$$
where $q,k,v\in R^{N_{2}\times \frac{d_{2}}{n_{2}}}$; $n_{2}$ is set to 12, LSA is the local self-attention calculation, LMSA is a local multi-head self-attention calculation with the same size as the input of the module $z_{m}^{l}$, and Softmax and Concat denote the same operations as in the global branch.

The LMSA calculation facilitates the comprehensive exploration of the correlations among the distinct local patches within each global patch. This type of local feature effectively captures the detailed semantic information in each local area of the image, thereby enabling the network to focus on meaningful local areas and compensating for the shortcomings of the basic Transformer backbone in extracting the internal information of global patches.

After completing the 12-layer encoding, we concatenate $N_{1}$ local patch embeddings $\left[z_{1}^{12},z_{2}^{12},\ldots ,z_{N_{1}}^{12}\right]$ to obtain the final output of the local branch, denoted as $f\in R^{N_{1}\times N_{2}\times d_{2}}$. The output f is fed into the fusion module along with the final output of the global branch $Z^{12}$.

### 3.3. Feature Fusion Module and Decoder

As previously indicated, we have extracted the long-range global contextual information within the whole image from the global branch and the local information within the patches from the local branch. Subsequently, the features derived from the two branches need to be combined by the feature fusion module so that the network can simultaneously focus on two extremely similar manipulation regions in the global context of the image and focus on the edge artifacts created by the copy–move forgery in the local regions. This would ensure that the focus regions are the copy–move forgery regions and that the region’s localization accuracy is higher.

Specifically, the global branch output $Z^{12}$ and the local branch output *f* are first reshaped as 2D feature maps $F_{g}\in R^{\frac{h}{p_{1}}\times \frac{w}{p_{1}}\times d_{1}}$ and $F_{l}\in R^{\frac{h}{p_{1}}\times \frac{w}{p_{1}}\times \left(N_{2}*d_{2}\right))}$, respectively. The two feature maps are then concatenated by the channels and fed into a convolution layer with a kernel size of 1 × 1, keeping the size of the output consistent with the two inputs, whereas the number of output channels is adjusted to 256. After the convolution is finished, a BatchNorm layer and a ReLu activation function are used to obtain the final output of the module $F_{\mathrm{gl}}$.

The design of the fusion module is quite simple, but it has proven to be effective. Since the IPRL designed in the local branch ensures that extraction of the local information is based on each global patch, and the position embedding has been added to every local patch to record the position information in the image while generating the local image block embedding, the global features and local features can be aligned in space. Therefore, only the channel-wise fusion needs to be considered in the fusion process. Mechanically, each channel of the feature map can be regarded as a feature descriptor. In this situation, a simple 1 × 1 convolution layer can be used for channel selection, eliminating extraneous description information and efficiently incorporating the local feature descriptors into the global features to refine the description of the dominant global features. The overall feature representation not only retains the judgment of two highly similar regions in the global feature, but it also assists with judging whether tampering traces actually exist in these suspected regions according to the local features and locates the edges of regions that have been tampered with by the semantic information within each local region more accurately. Figure 6 illustrates the specific structure of the feature fusion module.

After a 2D feature map with both global and local information $F_{\mathrm{gl}}$ is obtained, the feature map needs to be decoded to obtain the pixel-level predicted mask. We create a simple but useful decoder to up-sample the 2D feature map in a learnable way. The decoder also relies on the convolution layer as its fundamental module. Following the fusion of the two types of feature information by the feature fusion module, the output $F_{\mathrm{gl}}$ is fed into two 3 × 3 convolution layers. Each convolution layer also keeps the size of the outputs consistent with the input, and we implement a BatchNorm and ReLu layer afterwards. However, the first 3 × 3 convolution layer keeps the number of channels unchanged, whereas the second 3 × 3 convolution layer reduces the channel number to 1. Following the second 3 × 3 convolution layer, we up-sample the feature map by a factor of $p_{1}$ using a bilinear interpolation function to allow the final decoded mask $Y\in R^{h\times w\times 1}$ to match the spatial resolution of the original image. Thus far, Y is the predicted mask that indicates the position of the copy–move forgery occurring in the original image. Figure 6 illustrates the decoder’s specific structure.

### 3.4. Loss Function

In order to train LBRT, we use the binary cross-entropy loss (BCELoss) for localization tasks and fully supervise the predicted mask based on the pixel labels in the ground truth (GT) that are the same size as those of the original image, where a value of 0 represents the original pixel and a value of 1 represents a pixel that has been tampered with. BCELoss is formulated as
(8)BCELoss=−(ylog(Y)+(1−y)log(1−Y),
where y is the ground truth mask that indicates the position of the copy–move forgery. In addition, the LBRT solution adds a corresponding auxiliary decoding head, which takes the results from different layers of the Transformer encoder, inputs them into the fusion module and decoder, which are the same as those of the main task, and calculates the auxiliary losses by comparing the predicted mask with the GT, denoted as $aux_{i}\_loss$, where *i* represents the *i*-th encoder layer. The auxiliary losses are then summed to obtain the loss of the main task, denoted as decode_loss, to obtain the final total loss. The addition of an auxiliary loss has been shown to contribute to the convergence of model training [37]. The calculation of the total loss is formulated as
(9)$$Loss=decode\_loss+\sum aux_{i}\_loss,$$
where $aux_{i}\_loss$ represents the loss of the *i*-th auxiliary decoding head, decode_loss represents the loss of the main task, and Loss is the total loss. In this study, we take the 3rd, 6th, 9th and 12th layers of the Transformer encoder to feed into the auxiliary decoding head.

## 4. Experimental Results

The experiments were implemented within the PyTorch framework and MMSegmentation [38] on an NVIDIA RTX A6000 GPU. MMSegmentation is a code repository that is widely used for image segmentation tasks. Based on MMengine, which is a deep learning code repository developed by the OpenMMLab team, MMsegmentation provides a wide range of essential tools for training, testing, debugging, visualization, dataset management, and data augmentation.

### 4.1. Experimental Settings

**Datasets:** We conducted experiments on three public datasets for CMFD tasks that contain various copy–move-tampered images and corresponding ground truths that annotated the regions that have been tampered with. The regions related to copy–move forgery are marked in white, whereas the regions unrelated to copy–move forgery are marked in black. All datasets were split into training, validation, and testing sets in approximately an 8:1:1 ratio. The basic information of the three datasets is shown in Table 1.

USCISI is a synthetic CMFD dataset that was proposed by Wu et al. in [2] and consists of 100k samples, each with a binary ground truth that annotates the non-tampered with regions and regions that have been tampered with for CMFD. In our experiments, we randomly selected 80k, 10k, and 10k samples from the USCISI dataset for training, validation, and testing, respectively.

CASIA CMFD is a subset of the CASIA v2.0 [39] image tampering dataset for which Wu et al. manually verified 1313 copy–move forged images and created binary ground truths corresponding to these images in [2]. In our experiments, we randomly selected 1000, 133, and 180 samples from the CASIA CMFD dataset for fine-tuned training, validation, and testing, respectively, with a split ratio of approximately 8:1:1.

DEFACTO CMFD is a subset of the DEFACTO image tampering dataset proposed in [40]. We manually verified DEFACTO and selected 7057 copy–move tampered images with accurate annotations. We also reprocessed the binary ground truths corresponding to these images to improve their annotation accuracy, resulting in the DEFACTO CMFD dataset. In our experiments, we randomly selected 5645, 705, and 707 samples from the DEFACTO CMFD dataset for fine-tuned training, validation, and testing, respectively, with a split ratio of approximately 8:1:1.

**Baseline**: We selected six image forgery detection networks as baselines for comparison experiments with LBRT, including three networks specifically designed for CMFD tasks, namely BusterNet [2], CMSDNet [4], and UCM-Net [7], as well as three networks designed for all types of image content forgery, namely DenseFCN [6], MVSS-Net [24], and PSCC-Net [25]. All of these baselines are DCNN models, among which Busternet and DenseFCN are traditional DCNN models and the others are DCNN models with additional attention mechanisms.

**Metrics**: To quantify the localization performance, we report the precision, recall, F1-score, and accuracy (Acc), which are commonly used in the CMFD task, to assess the network’s performance in copy–move forgery detection and localization. The calculation of these metrics followed Strategy A mentioned in [2], where the precision, recall, and F1-score of the entire dataset are calculated after calculating the true positives (TPs), false positives (FPs), and false negatives (FNs) of all images in the test set.

**Implementation Details**: We conducted ablation experiments and comparative experiments to demonstrate the effectiveness of our proposed method. We trained LBRT and all of its variants as well as all the baseline models on the USCISI training set, which contained 80k copy–move tampered images. Then, we performed ablation experiments using the USCISI test set. For the comparative experiments, we tested all baselines on the three datasets mentioned in Table 1 and calculated the metrics. Specifically, for the experiments on the USCISI test set, we directly used the weights obtained from training on the USCISI training set. For the experiments on the CASIA CMFD and DEFACTO CMFD datasets, we fine-tuned the networks on the respective training sets of these two smaller datasets before testing and calculating the metrics on their test sets.

We used the ViT-base [26] weights pre-trained on ImageNet21K with an image resolution of 384 × 384 to initialize the backbone of LBRT’s global branch. During training, we followed the default settings of MMSegmentation, including the data augmentation and training schedules. We applied four data augmentation methods to the original images in the training and fine-tuned training sets: random resizing (ratio between 0.5 and 2), random cropping (cropped to 256 × 256), random horizontal flipping, and photometric distortion. The entire network was optimized using the SGD optimizer with momentum of 0.9 and weight decay of 0. The batch size was set to 8. On the USCISI training set, we set the initial learning rate to 0.01 and adjusted it using a polynomial learning rate decay schedule [37] with power of 0.9. The minimum learning rate was set to $1\times 10^{4}$, and the network was trained for a total of 100 epochs. On the CASIA CMFD and DEFACTO CMFD training sets, we fine-tuned the network with a learning rate of $1\times 10^{4}$. All original images input into LBRT and the other baselines were resized to 256 × 256.

### 4.2. Ablation Study

We designed two ablation experiments to validate the efficacy of incorporating the local refinement branch and using IPRL in the local refinement branch. We designed several variants as baselines for these two ablation experiments and evaluated them on the USCISI test set.

#### 4.2.1. Ablation Study of Dual-Branch Structure

The purpose of the first ablation experiment was to validate the impact of incorporating the local refinement branch on the detection of copy–move forgery. We designed several variants for this experiment. NA-LBRT only utilized the global branch, while TS-LBRT had both dual branches and a fusion module but trained the two branches separately in two stages: first, training the global branch and then freezing it, training the local branch, and finally fusing the feature extraction results of both branches using the fusion module. Finally, we used LBRT, which had both dual branches and the fusion module and trained both branches simultaneously. All baselines were evaluated for pixel-level localization on the USCISI test set. The detection results of the first ablation study are summarized in Table 2.

Based on the results observed from this ablation experiment, the following conclusions can be drawn.

The local refinement branch designed here effectively improves the performance of the model in the CMFD task. TS-LBRT outperforms NA-LBRT by 6.79% in terms of recall and 3.94% in terms of the F1-score. LBRT, which trains both branches simultaneously, has slightly lower precision (0.39% lower) than NA-LBRT but achieves 8.06% higher recall, leading to a 4.32% higher F1-score. This can be attributed to the dual-branch design, which not only finds the correlations between the source and target regions of copy–move forgery with a global perspective but also models the internal details of the global patches with a local perspective and injects these local details into the global information through the fusion module, resulting in more precise localization of tampered areas.Both the serial structure with staged training and the parallel structure with simultaneous training can improve the network’s performance in the CMFD task. However, the different network structures do not have a significant impact on the network’s performance. Observing the four metrics, TS-LBRT and LBRT show little difference in their values but outperform NA-LBRT significantly. Moreover, LBRT has a slight improvement over TS-LBRT. The result indicates that adding a local refinement branch can significantly improve the performance of the network in the CMFD task, with the advantage of simultaneous training in LBRT over the staged training in TS-LBRT. Moreover, LBRT can complete training at nearly twice the speed of TS-LBRT and has a simpler training process. Therefore, we define the simultaneous training structure of LBRT as the standard structure, and it served as the basis of the subsequent experiments.Adding a local branch only requires a minimal increase in the number of parameters, and the task of adding local information to LBRT does not require large-range pre-training. Regardless of whether staged training or simultaneous training is performed, only the initialization of the global branch needs pre-trained weights, whereas the local branch does not. This allows us to freely make improvements to the local branch. The results in Table 2 also demonstrate that adding the local branch only requires an increase of less than 2M parameters, which is very small.

#### 4.2.2. Ablation Study of IPRL

The purpose of the second ablation experiment was to verify the efficiency of the Intra-Patch Re-Dividing Layer (IPRL) design in the local refinement branch on the performance of CMFD. The ablation study was performed with three cases: (1) LBRT-NOSW, in which the IPRL was replaced with the Non-Overlapping Sliding Window (NOSW) described in [36]; (2) LBRT-CR, in which the IPRL was replaced by a simple re-dividing layer implemented as a 4 × 4 Conv layer (the local patch sequence was roughly reshaped to the size of (B∗256)×4×4×48 after 4 × 4 Conv); and (3) LBRT, the proposed LBRT model with the IPRL. We also designed several LBRT variants that used IPRLs with different local patch sizes, namely LBRT-8 × 8, LBRT, and LBRT-2 × 2 with local patch sizes of 8 × 8, 4 × 4, and 2 × 2, respectively, to compare with the variants of the different local block designs described above while exploring the effect of the local patch size on the network performance. NA-LBRT was also introduced as a control variant. All variants were evaluated for pixel-level localization on the USCISI test set. The detection results are summarized in Table 3.

Based on the results observed from this ablation experiment, the following conclusions can be drawn.

The design of the IPRL leads to the best performance in the CMFD pixel-level localization task. According to the results, all variants that utilize the local branch and re-dividing design outperform NA-LBRT by 4–5% in terms of the F1-score. Considering the variants with different re-dividing designs, including LBRT-NOSW, LBRT-CR, and LBRT (IPRL), the proposed LBRT with both the IPRL and 4 × 4 local patch size achieves the best overall performance, with the highest F1-score, recall, and accuracy among all variants in the second ablation study, whereas the number of parameters is only 87.76 M. Although LBRT-NOSW and LBRT-CR achieve higher precision than LBRT, their overall performance is still worse than that of LBRT. In addition, their corresponding re-dividing layers increase the number of parameters, especially with the design of the NOSW. The experimental results indicate that our Intra-Patch Re-Dividing Layer design enables us to extract local detail features for each global patch obtained from the whole image, and it realizes the effective addition of local information with only a limited increase in the number of network parameters. The design of the IPRL improves LBRT’s ability to extract local details and refines the detection results to achieve the highest performance with regard to detecting the copy–move regions that have been tampered with.The local patch size has little impact on the performance of both the local branch and the overall network. The metrics on the USCISI test set are close for the three patch sizes. Although the precision metric of LBRT, which indicates the proportion of pixels correctly detected as having been tampered with out of all the detected pixels, is slightly lower than that of the other variants, the overall performance is the best when the local patch size is set to 4 × 4, with the highest F1-score and accuracy among the three variants. The most noteworthy point is that although smaller patch sizes for the local patches would theoretically result in better performance, LBRT-2 × 2 does not show significantly improved performance compared to LBRT (with a 4 × 4 local patch size) and even shows slightly worse performance. This is due to the information duplication caused by the smaller patch size, which, while introducing more local information about regions that have been tampered with, also provides more local information describing other regions of the image. Consequently, when integrating global and local information, noise is introduced into the global features, which impairs the network’s ability to identify regions that have been tampered with.

Considering both the localization performance and the parameter size, we believe that a local patch size of 4 × 4 and the design of the IPRL constitute the most reasonable configuration for LBRT.

### 4.3. Comparison Experiment with State of the Art

The comparison experiments were divided into three parts that tested the results on three publicly available CMFD datasets: USCISI, CASIA CMFD, and DEFACTO CMFD. For the USCISI dataset, we directly tested the baselines trained on the USCISI training set on the USCISI test set. However, for the two smaller datasets, CASIA CMFD and DEFACTO CMFD, we further fine-tuned the baselines on the training subsets of these two datasets and then evaluated all baselines on the testing subsets of these two datasets. The fine-tuning aimed to mitigate the significant data distribution differences between the two smaller datasets and the larger dataset, USCISI.

Table 4 shows the pixel-level localization results of each baseline on the USCISI test set. The USCISI test set is highly consistent with the training data in terms of the data distribution, so all baselines achieved good localization performance. From the experimental results, it can be observed that LBRT outperforms all other baselines in most metrics. LBRT achieves a higher F1-score, precision, and accuracy compared to all other baselines, with only a slightly lower result for recall compared to PSCC-Net. In fact, PSCC-Net tends to identify more regions as regions that have been tampered with in the prediction of unknown data.

The above inference is supported by the visualization results in Figure 7. For example, in the third row, it is seen that PSCC-Net tends to identify two mobile phones as possible tampering source regions. It can also be observed that PSCC-Net detects more regions as regions that have potentially been tampered with, which occurs in a large number of images in the entire dataset. Whereas PSCC-Net occasionally identifies more regions that have been tampered with, it typically produces more false positives, which lowers its performance in terms of accuracy and F1-score, which reflect the overall performance. However, as shown in Figure 7, LBRT delivers more accurate detection results in more circumstances.

Table 5 presents the pixel-level localization results of each fine-tuned baseline on the CASIA CMFD test set. The CASIA CMFD dataset has significant differences in its data distribution compared to the USCISI dataset. It contains a large number of samples where the background regions are copied and moved with extreme similarity or without clear semantic information. It even includes many samples in which the copied regions overlap with the pasted regions. Such complex copy–move forgery samples are rarely seen in the USCISI dataset. Therefore, if the networks are trained only on the USCISI training set but are directly evaluated on the CASIA CMFD dataset without fine-tuning, the obtained metrics are usually low.

After fine-tuning with the same settings, LBRT outperforms all other baselines in all metrics. The F1-score of LBRT is 10.58% higher than that of UCM-Net, which is the next-best-performing network. The visualization results on the CASIA CMFD test set, shown in Figure 8, also demonstrate the superiority of the proposed method on this dataset. From Figure 8, it can be observed that, even after fine-tuning, the other baselines still produce many false detections in the complex CASIA CMFD dataset. For example, in the first row, copy–move forgery occurs on tree leaves that are difficult to notice, and only LBRT provides a roughly accurate localization of the forgery. In the fourth and fifth rows, where there are complex images with many other objects that are highly similar to the regions that have been tampered with, only LBRT can accurately identify the truly copied and moved objects.

Table 6 presents the pixel-level localization results of each fine-tuned baseline on the DEFACTO CMFD test set. DEFACTO CMFD also has notable differences in its data distribution, as it contains more complex samples that are not easily judged by the human eye, small target samples, and misleading samples compared to the USCISI dataset. For example, there are cases where small objects that have been tampered with appear alongside large objects that have not been tampered with but are visually similar, or objects that have been tampered with are mixed with many similar objects that have not been tampered with. However, compared to the CASIA CMFD dataset, DEFACTO CMFD has a slightly larger dataset size and the copy–move forgery images are more similar to those in the USCISI dataset. We also conducted fine-tuning with the same settings for each baseline on the DEFACTO CMFD dataset.

The result shows that LBRT significantly outperforms all other baselines in all metrics on this dataset. The F1-score of LBRT is nearly 11.40% higher than the second-ranked UCM-Net. This demonstrates that, even with different data distributions, as long as there is a sufficient amount of data for fine-tuning, LBRT can learn more comprehensive information related to copy–move forgery and achieve better prediction results. It can be observed in Figure 9 that LBRT can learn the similarities between regions that have been tampered with from a global perspective and the differences between the regions that have been tampered with from a local perspective in various copy–move forgery scenarios. By integrating these two features, LBRT can identify more correct regions that have been tampered with and achieve more accurate localization.

### 4.4. Visualization

Figure 10 illustrates the visualization results of the self-attention calculation process of LBRT. We visualize the self-attention maps of the global branch and the fusion self-attention maps of the two branches. The results show that the feature encoding process of the global branch allows LBRT to focus its attention on the pasted regions that have been tampered with and the corresponding copied regions. After adding the feature encoding process of the local branch, the attention weights that are dispersed in other irrelevant areas are reduced or even eliminated. The addition of the local branch enables LBRT to concentrate more on the regions that have been tampered with, especially regarding the details of the pasted target regions, leading LBRT to achieve more precise localization results in the final detection output.

## 5. Conclusions

We propose a local-information-refined dual-branch Transformer-based network for image copy–move forgery detection, called LBRT. In contrast to the current DCNN-based methods, the global context modeling branch of LBRT can exploit the fact that multi-head self-attention is adept at modeling global contexts to better search for remote dependent information from a global perspective, and it can partially replace the traditional calculation of feature self-correlations. Furthermore, the local refinement branch of LBRT serves to enhance the extraction of local information, where the basic Transformer encoder encounters difficulties. The local information in every local region can be completely extracted thanks to the design of the Intra-Patch Re-Dividing Layer on the global patches, and the feature fusion module ensures that the local information can be properly injected into the global information, increasing the global features’ attention to local details. Experiments showed that our network outperforms the state-of-the-art methods on the USCSI, CASIA CMFD, and DEFACTO CMFD datasets, indicating the great appropriateness and potential of LBRT for CMFD tasks. Our future efforts will focus on extending the current work to refine the network’s encoding and computational costs, making the network more lightweight and capable of differentiating between the sources and targets of copy–move forgery.

## Figures and Tables

**Figure 1 sensors-24-04143-f001:**
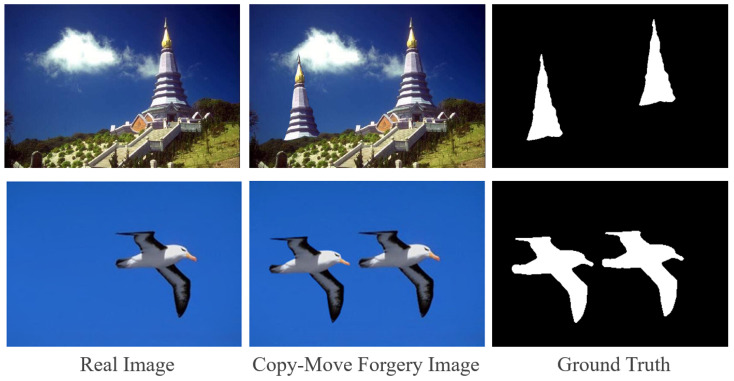
Examples of copy–move forgery images.

**Figure 2 sensors-24-04143-f002:**
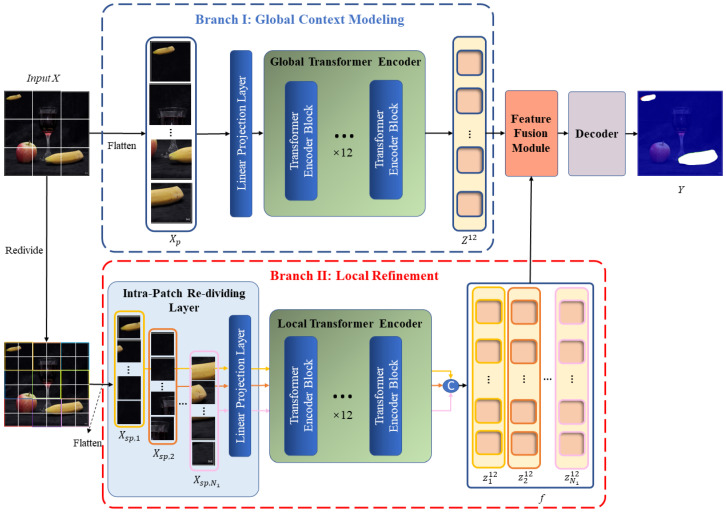
Architecture of the proposed LBRT.

**Figure 3 sensors-24-04143-f003:**
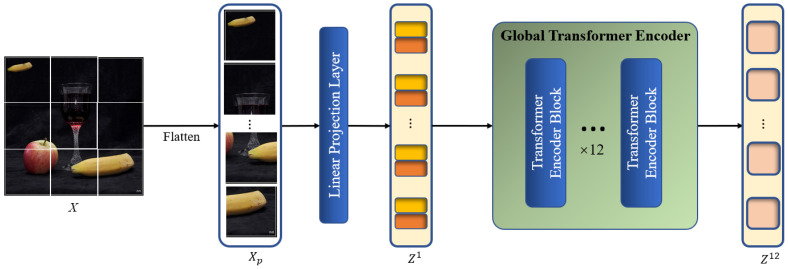
Architecture of the global context modeling branch.

**Figure 4 sensors-24-04143-f004:**
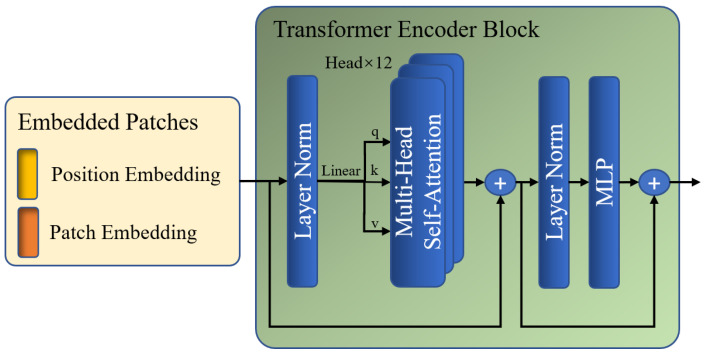
Architecture of the Transformer encoder block.

**Figure 5 sensors-24-04143-f005:**
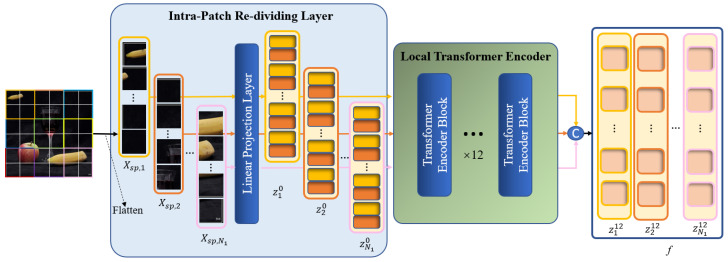
Architecture of the local refinement branch.

**Figure 6 sensors-24-04143-f006:**
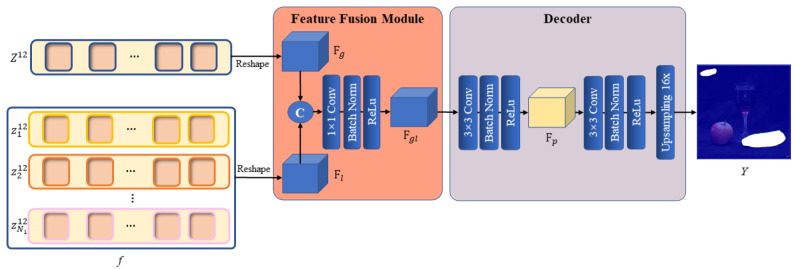
Architecture of the feature fusion module and decoder.

**Figure 7 sensors-24-04143-f007:**
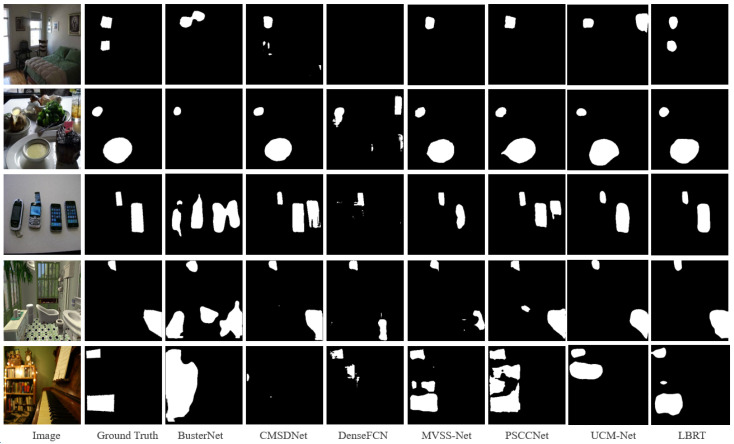
Visualization examples on the USCISI test set.

**Figure 8 sensors-24-04143-f008:**
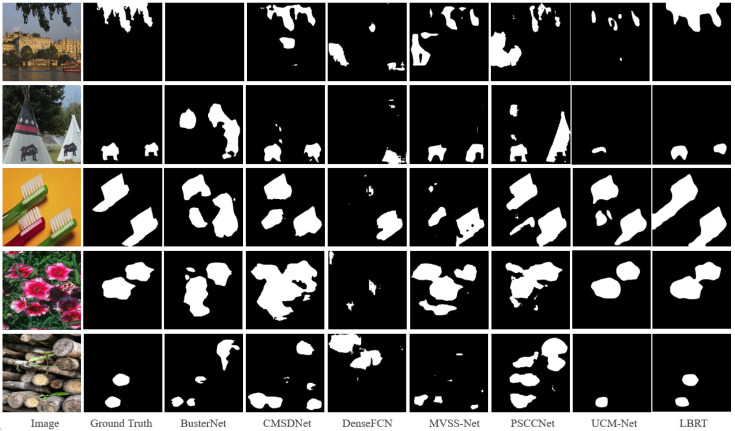
Visualization examples on the CASIA CMFD test set.

**Figure 9 sensors-24-04143-f009:**
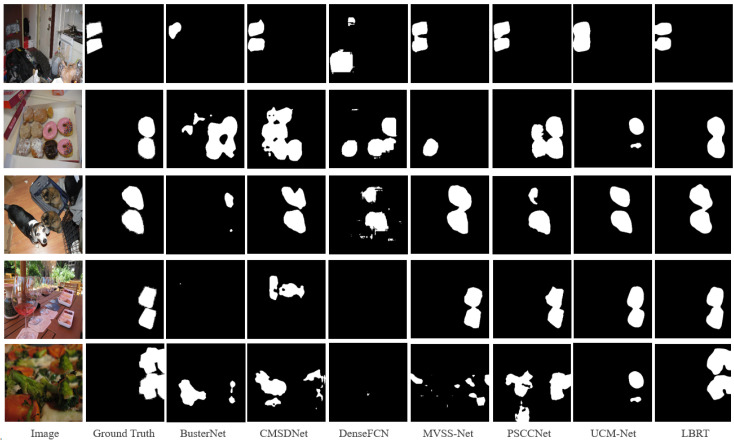
Visualization examples on the DEFACTO CMFD test set.

**Figure 10 sensors-24-04143-f010:**
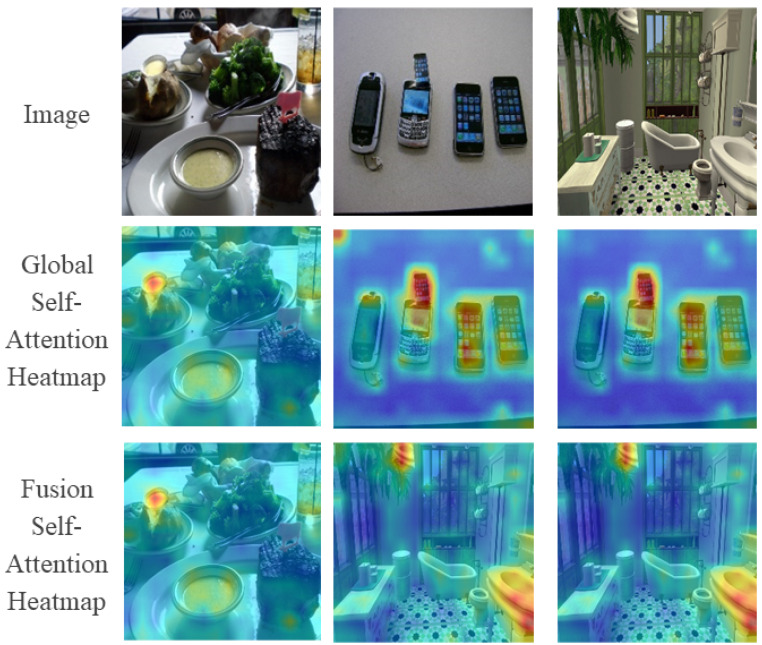
Visualization of self-attention heatmaps on three copy–move forgery images.

**Table 1 sensors-24-04143-t001:** Introduction to the datasets used.

Dataset	Total	Train/Validate/Test
USCISI [2]	100,000	80,000/10,000/10,000
CASIA CMFD [2]	1313	1000/133/180
DEFACTO CMFD	7057	5645/705/707

**Table 2 sensors-24-04143-t002:** Localization results of the ablation experiment with a dual-branch structure.

Method	Parameters	Flops	F1	Precision	Recall	Accuracy
NA-LBRT	86.04 M	23.27 G	79.74	86.94	73.64	96.00
TS-LBRT	87.76 M	25.14 G	83.68	**87.19**	80.43	96.65
LBRT (proposed)	87.76 M	25.14 G	**84.06**	86.55	**81.70**	**96.69**

**Table 3 sensors-24-04143-t003:** Localization results of the ablation experiment with the IPRL.

Method	Parameters	Flops	F1	Precision	Recall	Accuracy
NA-LBRT	86.04 M	23.27 G	79.74	86.94	73.64	96.00
LBRT-NOSW	175 M	116 G	83.87	**87.35**	80.66	96.68
LBRT-CR	88.35 M	25.30 G	83.45	86.80	80.35	96.61
LBRT-8 × 8	87.18 M	23.92 G	83.68	87.00	80.60	96.65
LBRT-2 × 2	90.12 M	30.91 G	83.51	86.57	80.65	96.61
LBRT (proposed)	87.76 M	25.14 G	**84.06**	86.55	**81.70**	**96.69**

**Table 4 sensors-24-04143-t004:** Localization results of the baselines on the USCISI test set.

Method	F1	Precision	Recall	Accuracy
BusterNet [2]	63.65	68.51	59.42	93.11
CMSDNet [4]	78.49	80.90	76.13	95.76
UCM-Net [7]	79.91	79.12	80.72	95.88
DenseFCN [6]	61.61	62.08	61.16	92.26
MVSS-Net [24]	81.87	82.23	81.51	96.33
PSCC-Net [25]	81.67	76.46	**87.65**	96.00
LBRT	**84.06**	**86.55**	81.70	**96.69**

**Table 5 sensors-24-04143-t005:** Localization results of the baselines on the CASIA CMFD test set.

Method	F1	Precision	Recall	Accuracy
BusterNet [2]	30.19	23.63	41.78	82.96
CMSDNet [4]	48.67	46.04	51.62	90.40
UCM-Net [7]	58.61	72.90	49.00	93.90
DenseFCN [6]	28.60	43.26	21.36	90.59
MVSS-Net [24]	44.16	57.56	35.82	92.01
PSCC-Net [25]	48.42	46.84	50.12	90.58
LBRT	**69.19**	**81.22**	**60.26**	**95.52**

**Table 6 sensors-24-04143-t006:** Localization results of the baselines on the DEFACTO CMFD test set.

Method	F1	Precision	Recall	Accuracy
BusterNet [2]	42.29	57.09	33.59	93.22
CMSDNet [4]	60.49	59.24	61.79	94.03
UCM-Net [7]	68.87	76.87	62.38	95.83
DenseFCN [6]	35.22	42.66	29.99	91.84
MVSS-Net [24]	64.10	72.95	57.17	95.26
PSCC-Net [25]	61.99	60.53	63.52	94.24
LBRT	**80.27**	**86.13**	**75.15**	**96.97**

## Data Availability

The USCISI dataset used in this study is openly available at https://github.com/isi-vista/BusterNet/tree/master/Data/USCISI-CMFD-Small (accessed on 23 June 2024). The CASIA dataset used in this study is openly available at https://www.kaggle.com/datasets/divg07/casia-20-image-tampering-detection-dataset (accessed on 23 June 2024). The DEFACTO dataset used in this study is openly available at https://www.kaggle.com/datasets/defactodataset/defactocopymove (accessed on 23 June 2024). The data generated and analyzed during the current study are available from the corresponding author on reasonable request.

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
