# Peer review of "LBRT: Local-Information-Refined Transformer for Image Copy–Move Forgery Detection"

_sensors, 2024, doi:10.3390/s24134143_

Round 1

Reviewer 1 Report

Comments and Suggestions for Authors

Contributions:

This study proposed a local information refined dual-branch Transformer-based network for image copy-move forgery detection. My comments are as follows:

  1. The full names of abbreviations should be provided when they first appear, such as LBRT USCI CASIA, CMFD, and DEFACTO. Please check them throughout this paper.
  2. The performance is acceptable.
  3. (Line 78 on page 3) Why is the abbreviation LBRT?
  4. (Page 6) Please briefly introduce the ViT-based standard.
  5. (Line 248 on page 7) The group number should be n1 rather than n.
  6. (Line 249 on page 7) Matrix A does not appear in eq. (2).
  7. (Page 7) What is the value of Dh?
  8. (Page 7) The symbols of matrices and vectors should be bolded.
  9. (Page 7) What are the dimensions of CSA(Zl) in eq. (2) and GMSA(Zl) in eq. (3)?
  10. (Line 254 on page 7) Are q, k, and v are matrices? If so, they are inconsistent with Fig. 2. They have been flattening.
  11. (Line 262 on page 7) Why do you use the symbol Z12?
  12. (Page 8) The borderline in Fig. 5 should be removed.
  13. (Line 285 on page 8) The statement should be rewritten.
  14. (Page 9) As shown in eq. (5), q and k are matrices. Why does it need a transpose operator on k?
  15. (Page 10) Equation (7) is unclear. Please define it explicitly.
  16. (Page 10) Section 4 needs to be renamed. It can be “experimental results”.
Comments on the Quality of English Language

The quality of the English language should be further improved.

Author Response

We feel great thanks for your professional review work on our article. As you are concerned, there are several problems that need to be addressed. According to your nice suggestions, we have made extensive corrections to our previous draft, the detailed corrections are listed below.

Comment 1: The full names of abbreviations should be provided when they first appear, such as LBRT USCI CASIA, CMFD, and DEFACTO. Please check them throughout this paper.

Response 1: Thank you for pointing this out. The full names of abbreviations has been added when they first appear in the main body of the manuscript. However, USCISI, CASIA and DEFACTO are the official names of these datasets, and we can not find their full names in their original papers, so we can not provide them.

Comment 2: The performance is acceptable.

Response 2: Thank you for your recognition of our work.

Comment 3: (Line 78 on page 3) Why is the abbreviation LBRT?

Response 3: Thank you for pointing this out.  The full name of LBRT is Local Branch Refinement Transformer. We have added LBRT's full name on page 3, line 81.

Comment 4: (Page 6) Please briefly introduce the ViT-based standard.

Response 4: Thanks for your suggestion. The ViT-base encoder uses a 12-layer encoding block, the patch size is 16x16, the number of heads in multi-head self-attention calculation is 12, and the number of channels of the patch embeddings is 768. The encoder of our global branch is designed in accordance with these hyperparameters. The above hyperparameters are represented correspondingly as l, p1, n1, and d1 in our manuscript. We have also added a detailed description of the ViT-base encoder on page 7, lines 236-245.

Comment 5:  (Line 248 on page 7) The group number should be n1 rather than n.

Response 5: We are really sorry for our careless mistakes. Thank you for your reminder and the mistake have been corrected in the revised version.

Comment 6:  (Line 249 on page 7) Matrix A does not appear in eq. (2).

Response 6: Thank you for pointing this out. We have modified formulas 2-3 and their descriptions on page 8 by adding the representation of matrix A.

Comment 7: (Page 7) What is the value of Dh?

Response 7: Thank you for pointing this out. The value of Dh is set to 256, which is equal to the length of the vectors q, k and v. We have added the description on page 8, lines 272-273.

Comment 8: (Page 7) The symbols of matrices and vectors should be bolded.

Response 8: Thank you for pointing this out. According to your suggestion, all of the symbols of matrices and vectors in Section 3 have been checked and bolded.

Comment 9: (Page 7) What are the dimensions of CSA(Zl) in eq. (2) and GMSA(Zl) in eq. (3)?

Response 9: Thank you for pointing this out. The dimensions of GSA(Zl) in eq. (2) and GMSA(Zl) in eq. (3) are both N1×d1, which are equal to the dimension of Zl. In our work, N1 and d1 represent the length of the patch embedding and the number of channels of the patch embedding respectively, which are set to 256 and 768 respectively.

Comment 10: (Line 254 on page 7) Are q, k, and v are matrices? If so, they are inconsistent with Fig. 2. They have been flattening.

Response 10: q, k, and v are two-dimensional matrices with length N1 and number of channels d1.  However, if we only consider its dimensions in space, they are actually obtained by flattening the color image in space, i.e. flattening from a two-dimensional image to a one-dimensional sequence.

Comment 11: (Line 262 on page 7) Why do you use the symbol Z12?

Response 11: The global patch embedding is denoted as Zl, where represents the l-th layer of the encoder. Therefore, Z12 means the final output of the golobal Transformer encoder after completing all 12 stacked layers. We have also added the relative description on page 8, line 290.

Comment 12: (Page 8) The borderline in Fig. 5 should be removed.

Response 12: Thank you for pointing this out. We have removed the borderline in Fig.4 and Fig.5 in the revised version.

Comment 13: (Line 285 on page 8) The statement should be rewritten.

Response 13: We are really sorry for our careless mistakes. Thank you for your reminder and the mistake have been corrected in the revised version.

Comment 14: (Page 9) As shown in eq. (5), q and k are matrices. Why does it need a transpose operator on k?

Response 14: As we mentioned in Response 10, q, k are two-dimensional matrices with length N1 and number of channels d1. When the matrix is multiplied, k needs to be transposed into a d1×N1 matrix, so that each channel of q can be multiplied and summed with each channel of k, and finally the attention matrix A with the dimension of N1×N1 is obtained.

Comment 15: (Page 10) Equation (7) is unclear. Please define it explicitly.

Response 15: Thank you for pointing this out. In the revised version, the equation is numbered 9, and we have added euqation (8) to the manuscipt, along with a detailed description of euqation (9) to clearly define the formula.

Comment 16: (Page 10) Section 4 needs to be renamed. It can be “experimental results”.

Response 16: Thanks for your suggestion. We have modified the name of Section 4 in the revised version.

Comment 17: The quality of the English language should be further improved.

Response 17: Thanks for your suggestion. We feel sorry for our poor writings. We have used the paid editing service of MDPI to help us polish the language in the revised manuscript. We hope the revised manuscript could be acceptable for you.

Reviewer 2 Report

Comments and Suggestions for Authors

1. Innovation and Originality

  • Novel CMFD Network: The introduction of a Transformer-based CMFD network with a dual-branch structure to capture both global and local information is innovative. This approach addresses the limitations of previous DCNN-based methods by leveraging the strengths of Transformers in modeling global context and refining local details.
  • Local Information Refinement: The refinement of Transformer encoding in the local branch and the implementation of the Intra-Patch Re-dividing Layer (IPRL) demonstrate an original approach to enhancing local feature extraction without significantly increasing model complexity.

2. Technical Contribution

  • Dual-Branch Structure: The use of a dual-branch structure that separately processes global and local information before fusing them is a significant technical contribution. This method ensures a more comprehensive feature extraction process, potentially leading to more accurate forgery detection.
  • Intra-Patch Re-dividing Layer (IPRL): The IPRL is a novel layer that re-divides global patches into smaller local patches, ensuring detailed local feature extraction. This innovation balances the model's complexity and performance, enhancing local detail extraction with minimal parameter increase.

3. Methodology

  • Model Description: The proposal should provide detailed descriptions of the global context modeling branch and the local refinement branch, including how the multi-head self-attention mechanism is applied and how the IPRL operates.
  • Feature Fusion: The process of fusing features from the global and local branches should be clearly explained, detailing how the integration enhances the overall feature representation.

4. Experimental Validation

  • Datasets: The use of multiple datasets (USCISI, CASIA CMFD, and DEFACTO CMFD) for validation is a strong point, providing comprehensive evaluation across different scenarios and image types.
  • Comparative Analysis: The proposal claims that the LBRT outperforms advanced techniques, including traditional DCNN models and those with additional attention mechanisms. Specific performance metrics, results, and comparisons should be included to substantiate this claim.

5. Practical Implications

  • Improvement in Accuracy: By refining global features with local edge information, the proposed method improves the accuracy of locating copy-move forgery regions, which is a critical aspect of CMFD.
  • Model Efficiency: The minor increase in model parameters due to the local branch refinement suggests that the proposed method is efficient, making it practical for real-world applications without the need for extensive computational resources.

6. Future Work

  • Refinement and Optimization: The future work should focus on refining the network's encoding processes and optimizing computational costs, ensuring that the model remains lightweight and efficient.
  • Source and Target Differentiation: Enhancing the model's capability to differentiate between the source and target of copy-move forgery will further improve its practical applicability and robustness.

Author Response

Thank you very much for taking the time to review this manuscript. According to your nice suggestions, we have made extensive corrections to our previous draft, the detailed corrections are listed below.

Comment 1:

Innovation and Originality

  • Novel CMFD Network: The introduction of a Transformer-based CMFD network with a dual-branch structure to capture both global and local information is innovative. This approach addresses the limitations of previous DCNN-based methods by leveraging the strengths of Transformers in modeling global context and refining local details.
  • Local Information Refinement: The refinement of Transformer encoding in the local branch and the implementation of the Intra-Patch Re-dividing Layer (IPRL) demonstrate an original approach to enhancing local feature extraction without significantly increasing model complexity.

Response 1:

Thank you for your nice comments on our article. We appreciate to your recognition of the innovation and originality in our work. 

Comment 2:

Technical Contribution

  • Dual-Branch Structure: The use of a dual-branch structure that separately processes global and local information before fusing them is a significant technical contribution. This method ensures a more comprehensive feature extraction process, potentially leading to more accurate forgery detection.
  • Intra-Patch Re-dividing Layer (IPRL): The IPRL is a novel layer that re-divides global patches into smaller local patches, ensuring detailed local feature extraction. This innovation balances the model's complexity and performance, enhancing local detail extraction with minimal parameter increase.

Response 2:

Thank you for your nice comments on our article. We appreciate to your recognition of the technical contribution in our work. 

Comment 3:

Methodology

  • Model Description: The proposal should provide detailed descriptions of the global context modeling branch and the local refinement branch, including how the multi-head self-attention mechanism is applied and how the IPRL operates.
  • Feature Fusion: The process of fusing features from the global and local branches should be clearly explained, detailing how the integration enhances the overall feature representation.

Response 3:

Thank you for pointing this out. We agree with the comment about model description and feature fusion. In response to your comments, we have revised the descriptions of global branches, local branches, and feature fusion module in sections 3.1 to 3.3, respectively. We have added a detailed explanation of how multi-head self-attention works in global branches on lines 277-283 on page 8. We have added a description about how IPRL works in local branch and the mechanism of local multi-head self-attention in lines 320-325 on page 9 and 335-340 on page 10, respectively. We have revised and refined the description of how the feature fusion module fuses global and local features and the explanation of why it fully considers the two features and enhances the overall feature representation well on page 10, lines 360 -374.

Comment 4:

4. Experimental Validation

  • Datasets: The use of multiple datasets (USCISI, CASIA CMFD, and DEFACTO CMFD) for validation is a strong point, providing comprehensive evaluation across different scenarios and image types.
  • Comparative Analysis: The proposal claims that the LBRT outperforms advanced techniques, including traditional DCNN models and those with additional attention mechanisms. Specific performance metrics, results, and comparisons should be included to substantiate this claim.

Response 4:

Thank you for pointing this out. We agree with the comment about comparative analysis, but the models compared to LBRT in this section  have already included both traditional DCNN models and models with additional attention mechanism. We have accordingly modified the description of each baseline on page 12, lines 437-443, emphasizing which models are traditional DCNN models and which models have additional attention mechanism.

Comment 5:

Practical Implications

  • Improvement in Accuracy: By refining global features with local edge information, the proposed method improves the accuracy of locating copy-move forgery regions, which is a critical aspect of CMFD.
  • Model Efficiency: The minor increase in model parameters due to the local branch refinement suggests that the proposed method is efficient, making it practical for real-world applications without the need for extensive computational resources.

Respons 5:

Thank you for your nice comments on our article. We appreciate to your recognition of the practical implications in our work. 

Comment 6:

Future Work

  • Refinement and Optimization: The future work should focus on refining the network's encoding processes and optimizing computational costs, ensuring that the model remains lightweight and efficient.
  • Source and Target Differentiation: Enhancing the model's capability to differentiate between the source and target of copy-move forgery will further improve its practical applicability and robustness.

Respons 6:

Thank you again for your positive comments and valuable suggestions to improve the quality of our manuscript. We will consider focusing our future work on these aspects according to your suggestions.

Reviewer 3 Report

Comments and Suggestions for Authors

1.The structure of the abstract is generally clear, but some sentences lack smooth transitions. It is suggested to begin the introduction of the LBRT model with a brief explanation of how TNT and EDTER inspired this model, followed by a detailed description of LBRT's innovations.
1. 摘要的结构一般是清晰的,但有些句子缺乏平滑的过渡。建议开始介绍LBRT模型,简要说明TNT和EDTER如何启发这种模型,然后详细描述LBRT的创新。

2. The internal logic within each section is clear, gradually guiding readers to understand the design and implementation of the LBRT model. However, there is a lack of transitional sentences and conjunctions to enhance coherence between paragraphs.
2.每一部分的内在逻辑清晰,逐步引导读者理解LBRT模式的设计和实施。然而,缺乏过渡句和连接词来加强段落之间的连贯性。

3. The description of the shortcomings of existing methods is somewhat brief. It is recommended to elaborate on the specific deficiencies of deep convolutional neural networks and Transformer structures in CMFD, to enhance readers' understanding of the necessity of the research. Detailed descriptions of the innovative features and advantages of the LBRT model can increase readers' interest and appreciation of this approach.
3.对现有方法的缺点的描述有些简短。建议详细阐述CMFD中深度卷积神经网络和Transformer结构的具体不足,以增强读者对研究必要性的理解。详细介绍LBRT模式的创新特点和优势,可以增加读者对这种方法的兴趣和欣赏。

4. The description of the "Local Refinement Branch" is quite detailed, but it is suggested to provide more detailed explanations on how the "global branch" and "local refinement branch" interact with each other.
4.对“局部细化分支”的描述相当详细,但建议对“全局分支”和“局部细化分支”如何相互作用进行更详细的说明。

5. For the "ViT-base encoder with dual-branch design" section, it is recommended to provide a detailed explanation of why the ViT-base encoder was chosen and its advantages in CMFD tasks. For the "feature extraction process" section, detailed explanations of the specific implementation methods and interactions between the global context modeling branch and the local refinement branch are suggested. Regarding the "fusion module" section, it is advisable to provide detailed explanations of the specific methods and advantages of feature fusion.
5.在“采用双分支设计的ViT基编码器”部分,建议详细解释为什么选择ViT基编码器及其在CMFD任务中的优势。对于“特征提取过程”部分,建议详细解释全局上下文建模分支和局部细化分支之间的具体实现方法和交互。关于“融合模块”部分,宜对特征融合的具体方法和优点进行详细说明。

Author Response

We feel great thanks for your professional review work on our article. As you are concerned, there are several problems that need to be addressed. According to your nice suggestions, we have made extensive corrections to our previous draft, the detailed corrections are listed below.

Comment 1:

The structure of the abstract is generally clear, but some sentences lack smooth transitions. It is suggested to begin the introduction of the LBRT model with a brief explanation of how TNT and EDTER inspired this model, followed by a detailed description of LBRT's innovations.

Response 1:

Thank you for your nice comments on our article. According to your suggestions, we have checked the content of the abstract and adjusted some of the statements. We have modified the description of TNT and EDTER in lines 194-208 on page 5, and briefly explained which contents of these two works inspired our proposed model in lines 210-218. In addition, we have added a description of some LBRT innovations in sections 3.1 to 3.3, which can be found in paragraph 1 of section 3.1 on page 7, paragraph 1 of section 3.2 on page 8, and paragraph 3 of section 3.3 on page 10.

Comment 2:

The internal logic within each section is clear, gradually guiding readers to understand the design and implementation of the LBRT model. However, there is a lack of transitional sentences and conjunctions to enhance coherence between paragraphs.

Response 2:

Thank you again for your positive comments and valuable suggestions to improve the quality of our manuscript. We edited the entire manuscript in English, believing that these changes would make the transition between paragraphs more coherent.

Comment 3:

The description of the shortcomings of existing methods is somewhat brief. It is recommended to elaborate on the specific deficiencies of deep convolutional neural networks and Transformer structures in CMFD, to enhance readers' understanding of the necessity of the research. Detailed descriptions of the innovative features and advantages of the LBRT model can increase readers' interest and appreciation of this approach.

Response 3:

Thank you for your suggestions on our article. In two paragraphs on page 2, we have elaborated on the specific shortcomings of the existing DCNN models and approaches with Transformer structure in CMFD. The introduction of the innovative features and benefits of LBRT is summarized on page 3, and some details are added when the specific content of the method is introduced in Methodology. If there are any other modifications we could make, we would like very much to modify them and we really appreciate your help.

Comment 4:

The description of the "Local Refinement Branch" is quite detailed, but it is suggested to provide more detailed explanations on how the "global branch" and "local refinement branch" interact with each other.

Response 4:

Thanks for your suggestion. At the stage of feature extraction, the two branches are completely parallel, and the interaction between them is reflected in the comprehensive consideration and fusion of the two features at the stage of feature fusion. According to your suggestions, we have provided a more detailed explanation about the fusion mechanism of the two branches and how they contribute to the overall feature representation together on pages 360-374.

Comment 5:

For the "ViT-base encoder with dual-branch design" section, it is recommended to provide a detailed explanation of why the ViT-base encoder was chosen and its advantages in CMFD tasks. For the "feature extraction process" section, detailed explanations of the specific implementation methods and interactions between the global context modeling branch and the local refinement branch are suggested. Regarding the "fusion module" section, it is advisable to provide detailed explanations of the specific methods and advantages of feature fusion.

Response 5:

Thank you for your nice comments on our article. According to your suggestions, we have added a detailed explanation of the reasons for choosing ViT-base encoder and its advantages in CMFD tasks on pages 6-7, lines 233-245. As we mentioned in Response 4, the two branches run in parallel in the feature extraction process, and their interaction needs to be reflected in the fusion module. For comments about the feature fusion section, we have added a detailed description of the feature fusion process and its advantages on page 345-374 of page 10.

Thank you again for your positive comments and valuable suggestions to improve the quality of our manuscript.

Reviewer 4 Report

Comments and Suggestions for Authors

The authors describe an approach for detecting and locating copy-move forgery based on a dual-branch transformer structure allowing to extract local and global information. It performs Transformer encoding on global patches divided from the image and local patches re-divided from global patches using a global modeling branch and a local refinement branch, respectively. The self-attention features from both branches are combined, up-sampled, and decoded. Empirical results show the quality of the proposal. Overall, the paper is nicely written, well motivated, the proposal is well described. The manuscript provides a nice contribution to the litterature.

Author Response

Comments 1: The authors describe an approach for detecting and locating copy-move forgery based on a dual-branch transformer structure allowing to extract local and global information. It performs Transformer encoding on global patches divided from the image and local patches re-divided from global patches using a global modeling branch and a local refinement branch, respectively. The self-attention features from both branches are combined, up-sampled, and decoded. Empirical results show the quality of the proposal. Overall, the paper is nicely written, well motivated, the proposal is well described. The manuscript provides a nice contribution to the litterature.

Response 1: Thank you very much for taking the time to review this manuscript. We appreciate your recognition of our work. If there are any other modifications we could make, we would like very much to modify them and we really appreciate your helps.

Round 2

Reviewer 1 Report

Comments and Suggestions for Authors

The authors have improved the quality of this paper. I think this paper can be accepted for publication.

Comments on the Quality of English Language

The quality of the Ehglish language is acceptable.